# The Burden of Survivors: How Can Phage Infection Impact Non-Infected Bacteria?

**DOI:** 10.3390/ijms24032733

**Published:** 2023-02-01

**Authors:** Andrey V. Letarov, Maria A. Letarova

**Affiliations:** Winogradsky Institute of Microbiology, RC Biotechnology RAS, 119991 Moscow, Russia

**Keywords:** bacteriophage ecology, intercell communication of bacteria, phage-host interactions

## Abstract

The contemporary understanding of complex interactions in natural microbial communities and the numerous mechanisms of bacterial communication challenge the classical concept of bacteria as unicellular organisms. Microbial populations, especially those in densely populated habitats, appear to behave cooperatively, coordinating their reactions in response to different stimuli and behaving as a quasi-tissue. The reaction of such systems to viral infection is likely to go beyond each cell or species tackling the phage attack independently. Bacteriophage infection of a fraction of the microbial community may also exert an influence on the physiological state and/or phenotypic features of those cells that have not yet had direct contact with the virus or are even intrinsically unable to become infected by the particular virus. These effects may be mediated by sensing the chemical signals released by lysing or by infected cells as well as by more indirect mechanisms.

## 1. Introduction

The bacteriophages discovered more than 100 years ago by F. Twort [1] and F. D’Herelle [2] became a valuable model for investigating the nature of viruses and studying the basic molecular mechanisms of life. Bacteriophage research was the main approach that eventually led to the currently established concept of the virus as a transmissive genetic program [3]. Today, bacteriophage biology has developed into one of the most elaborate fields of virology, which is able to establish the link between the phenomena observed at all levels of the biological organization ranging from the sub-molecular to the biosphere. Initially, bacteriophages were viewed mostly as model objects for discovering general mechanisms of life. Now, they are considered a major force shaping microbial landscapes in most natural habitats [4,5,6]. Our understanding of phage and bacteria relationships in nature as being solely host-parasite or predator-prey connections is being replaced by the concept of multifaceted interactions. These interactions often may be considered as mutualistic according to their outcomes for both bacterial and viral populations [7,8,9]. Based on the perspective of contemporary knowledge, bacteriophages may therefore be called “natural frenemies” rather than “natural enemies” of bacteria.

Interestingly, the bacteriophage discovery took place during an era when the conceptualization of a microorganism by microbiologists was shifting from bacterial culture to bacterial cell as a representation of an individual organism [10]. Currently, this concept is again undergoing a significant transformation, yielding to the view of microbial populations as (quasi) multicellular organisms [11,12,13,14,15,16]. We understand dense bacterial populations—such as biofilms, bacterial filaments, swarms, fruiting bodies or even concentrated planktonic suspensions in laboratory cultures—as organized multicellular systems or microbial tissues. This raises the question whether the general mechanisms underlying the reaction to the viral infection are as widely distributed in such systems as they are in the tissues of true multicellular animals or plants. 

The highly selective bactericidal action of bacteriophages led F. d’Herelle to the idea of harnessing them to combat bacterial infections [17,18]. Although phage therapy saw a steep decline after the 1940s, over the last 20 years interest in the antimicrobial activity of bacteriophages has experienced a renaissance in connection with the global problem of resistance of microbial pathogens to antibiotics [19,20,21,22]. Different phage-based technologies for the biocontrol of undesirable bacterial populations in industry or the environment have also been developed [23,24,25]. In natural habitats, however, bacteriophages tend to establish a long-term co-existence with sensitive bacterial hosts [7,26,27] rather than driving the host population to extinction. Accordingly, using phages for therapy or biocontrol requires application strategies that allow mass killing of the target bacterium while preventing viruses from showing their natural inclination to “make a deal” with the host population. This calls for studying the entire spectrum of possible interactions between phage and bacterial populations.

Until recently the research in this field was mostly focused on aspects such as highly specific and population-density-dependent lytic action of phages, their contribution to lateral gene transfer, and mechanisms and outcomes of the co-evolution of phages and their hosts in experimental systems or natural habitats. A less obvious array of phenomena regarding the effect of phage infection on those fractions of bacterial populations that do not (or have not yet) directly interacted with the phage has been largely overlooked. Phage infection may influence the uninfected bacteria based on the physiological activity of the products secreted by infected cells (virocells) and/or released products of bacterial lysis. Recipients of such an exposure can be the cells of the same species or strain, potentially sensitive to the phage, as well as other components of the microbial community, namely those that are resistant to the action of that particular bacteriophage. We refer below to such phenomena as the “physiology of survivors effects” (POSE). The POSE-causing agents can be specialized bacterial signal-transmitting molecules, for example quorum-sensing effectors. They can also include (1) signaling molecules whose synthesis is encoded by the genome of the phages, (2) nanometer-scale complexes, including the phage particles themselves (beyond their direct infectious action), (3) membrane vesicles of the host bacterium or (4) low molecular weight metabolites of lysed cells which can also cause specific reactions of the non-infected cells, especially in dense microbial communities such as biofilms, or in laboratory cultures.

The effects arising from the shift in the composition of the surviving part of bacterial populations as a result of selection by phage infection pressure can formally also be considered as POSE. This type of phenomena has been studied extensively (see the reviews [28,29], also book [30]). In this review we concentrate on the known examples of POSEs mediated by different types of signals generated by phage infection.

## 2. POSEs Mediated by Phage Particles or Phage Structural Proteins

### 2.1. Phage Infection Influencing the Microbial Community’s Spatial Organization

The formation of a halo around plaques formed by some bacteriophages was one of the first described POSE mediated by bacteriophage structural proteins. The halo-forming phages encode polysaccharide-degrading enzymes, which help them to penetrate the capsules and/or exopolysaccharides produced by their hosts. Most of the known phage polysaccharide depolymerases are phage structural proteins such as tail spikes. During the infection these proteins may be produced in significant excess, spreading out of the viral infection site (e.g., the plaque) faster than phage particles. This modifies the surface of the cells in the non-infected area of the bacterial lawn, adjacent to the plaques. Removing the capsules in the context of natural environments may alter interactions of the bacteria to various external factors. For example, capsule-degrading enzymes were shown to lift the protection of microbial cells against immunity factors mediating the in vivo therapeutical effect of phage-derived tail spikes which lack any direct bactericidal activity in vitro [31,32]. The enzymes produced by the phage-infected cells in the course of phage therapy (PT) may thus contribute to the elimination of the non-infected pathogen cells by the immune system.

The disruption of the biofilm structure by bacteriophage-encoded enzymes [33] may also contribute to PT efficiency as well as cause POSEs in natural ecosystems. As a proof of this principle T7 phage was engineered to express DspB in infected cells. Being released during the cell lysis this enzyme degrades the exopolysaccharide (polymeric β-1,6-N-acetyl-d-glucosamine) produced by *E. coli* TG1 strain. The treatment of the pre-grown *E. coli* TG1 biofilm with the engineered phage yielded 100 times greater biofilm reduction compared to the non-modified virus [34]. Clearly, degrading the matrix of a multi-species biofilm based on the phage infection of only one of microbial species may have strong and multifaceted impact on the co-habitant species that are not sensitive to this phage. The physiological state of bacteria in a biofilm is often different from the physiological pattern of planktonic cells (see [35,36] for reviews). Breaking the biofilm down may rapidly alter bacteria metabolic activity, partly due to cell dispersal and a decrease in the quorum-sensing (QS) signal(s) concentration. Moreover, the polysaccharide degradation products (oligosaccharides) may be sensed by bacteria and act as a secondary signal triggering certain alterations in the gene expression patterns and/or adhesion properties of surrounding bacteria. To our knowledge, the oligosaccharide reception by bacteria has not yet been investigated, and the evidence for it is very scarce and indirect [37,38,39]. At the same time a vast amount of data is available on the physiological activity of such molecules in animals and plants (see reviews [40,41]), suggesting that bacteria may share similar mechanisms. Finally, in the case of symbiotic and pathogenic bacteria, the influence of oligosaccharides produced by phage-encoded enzymes on the macro-hosts’ systemic or local reactions may influence the fate of the microbial community.

Biofilm disruption due to the action of phage-encoded enzymes is merely one specific example of a class of POSE relying on a modification of spatial structure of microbial communities. Temperate filamentous Pf4-like bacteriophages of *Pseudomonas aeruginosa* are involved in various processes during the biofilm development by their host, including the formation of the correct biofilm spatial organization. During the initial stages of biofilm development by the PAO1 strain lysogenic for Pf4, the so-called superinfectious variants (mutants) of the phage emerge in the growing microcolonies settled on the surface colonized by bacteria. These superinfectious variants are capable of infecting cells, lysogenically by Pf4 phage. Nonetheless, they act in a selective manner, mediating the death of some cells in the middle of the forming microcolonies but leaving the cells at the microcolony periphery uninfected. Such selectivity is based on different expression of the pili serving the Pf4 receptors. The spatially controlled death of a part of the microbial population yields biofilm channels [42,43]. The surviving bacterial cells, and possibly other cells in multi-species biofilms, benefit from the more robust biofilm following the optimal spatial design [42].

A specific spatial structure may be also imposed on the community by the pattern of lysis/lysogeny decisions in temperate phages. Some phages were reported to produce plaques with regular patterns of concentric rings with higher and lower density of the lysogens’ microcolonies [44,45,46]. The mechanism of the concentric ring formation is most probably based on a diffusible signal(s) produced by the cells that undergo lytic or lysogenic infection. These signals are sensed by other cells to alter the probability of lysogenization upon contact with the phage. The nature of this signal has not yet been elucidated in any such system. The phage particles themselves can potentially serve as a “signal” controlling the lysogenization frequency. The probability of the lysogenic decision increases substantially in those cells simultaneously infected by multiple particles of the lambdoid phages [47,48,49]. The likely explanation is a higher gene dosage of CII, the key transcription factor activating the lysogenic pathway. Since the degradation of the CII protein depends on a amount of Hfl (FtsH) protease present in the cell at the time of infection, an increased number of cII gene copies facilitates a more rapid accumulation of the CII concentration necessary for the lysogenic decision [47,50]. A similar mechanism may be expected in many other phages in which the lysogeny control is similar to that of the phage lambda [48,50]. Note that mathematical modeling by Mitarai et al. [45] predicts no significant alterations of the lambda plaque upon switching the effect of the multiplicity of infection. Nonetheless, this mechanism still cannot be reliably ruled out. The other signals that potentially control the lysogenization frequency (metabolites, small RNAs, membrane vesicles) are considered below. Regardless of the mechanism behind the concentric pattern formation, if phage-insensitive microorganisms are present in the system, then such a pattern should create for them the alternating zones of higher and lower intensity of the competition with the lysogens formed from the phage-sensitive strain. (This assumes that the density of the sensitive cells escaping from infection and/or phage-resistant mutants is much lower).

Even without a special, sophisticated lysis/lysogeny control in space, phage infection may locally and/or temporally decrease the population of sensitive cells. This, in turn, creates an opportunity for the other strains and species to colonize the niche. Both mathematical and experimental modeling have shown the increased ability of the lysogenic bacteria to invade the environmental niches densely populated by those bacteria sensitive to the corresponding phage [51,52]. Although this phenomenon is beyond the scope of the present review (because the lysogens that release the phages were formed as a result of direct infection), quite a similar *bona fide* POSE mechanism was recently described for some motile bacteria (*Pseudomonas putida*, *Viridibacillus* sp., *Enterobacter* sp., *Serratia* sp., *Bacillus* sp., *Janthinobacterium* sp.) that are able to non-specifically adsorb bacteriophages infective for other species, such as *E. coli* [53]. Such phage-loaded bacteria may transport viruses over considerable distances. Interestingly, some bacterial species were found to be able to travel and to transport phages along the hyphae of fungi, increasing their spread in viscous media such as soft agar in the laboratory model. The phages liberated from the carrier cells’ surface may cause local lysis of the resident population (e.g., formation of a plaque on the sensitive strain lawn). This makes room for colony formation by the invading species. Note, however, that the same bacteria, even when assisted by the fungal mycelium, were unable to establish colonies in the absence of the phage: they were outcompeted by the dense and rapidly growing background population (*E. coli* lawn). This novel trait of an increased ability for invasion acquired by bacteria insensitive to the bacteriophage (as a result of the phage infection of different bacterial species) exactly matches our definition of POSE.

### 2.2. Phage Virions and Phage Proteins Serving as Structural Components of Bacterial Cells and Biofilms

The bacteriophage proteins released from infected cells may bind to the surface of non-infected cells (or non-induced lysogens) of the same or different species and to functionally mimick the own bacterial surface proteins that are delivered to the cell surface *via* secretion pathways. The proteins PblA and PblB of the temperate bacteriophage SF100 *Streptococcus mitis* are released by the lysis of a fraction of the cells after prophage induction. Those proteins bind to the surface of the remaining cells and serve as adhesins for attachment to the platelet surface. This plays a role in the pathogenesis of endocarditis caused by *S. mitis* [54,55]. The endolysin of this phage, by a similar mechanism, also played the role of adhesin, mediating the interaction of *Streptococcus* cells with fibrinogen [56]. These phenomena raise a question: to what extent does the acquisition of the surface proteins from outside (not necessarily due to a nearby phage infection) contribute to shaping the molecular landscape of microbial cells in natural habitats.

As discussed above, the whole phage virions also may bind bacterial cells without infecting them. Such surface-exposed virions potentially convey some new features to bacteria, for example by serving as adhesins. Currently, however, we know of no such examples in which whole-virions function as a decoration of bacterial cells, the involvement of some phages into building of such complex structure as the extracellular matrix of microbial biofilms. The virus particles of the above-mentioned temperate *P. aeruginosa* filamentous bacteriophage Pf4, produced upon prophage activation, are released from the cells in a process that is not associated with cell death (for details of filamentous phage morphogenesis see reviews [57,58]). Virions interact with external cell polysaccharides and other biopolymers including extracellular DNA. This yields liquid crystalline complexes that can assemble into droplets occluding individual *P. aeruginosa* cells, reducing their sensitivity to antibiotics [59]. Importantly, these droplets were able to assemble around inanimate, rod-shaped particles. This indicates that the assembly of the phage liquid crystalline shell depends more on the size and shape of the particles than on specific features of the cell surface. Therefore, other bacteria located near a lysogenic *P. aeruginosa* population may acquire protective structures formed by Pf4-related phages.

The liquid crystals made of Pf virions may also be incorporated into larger structures contributing to the *P. aeruginosa* biofilm matrix. These structures increase the biofilm’s mechanical strength and improve its desiccation resistance ([60], see also review [61]). Earlier studies reported the participation of unidentified tailed bacteriophage particles in forming dental plaque biofilms, serving as a structural component of the matrix (see review [62]). These phage particles can be highly abundant in the material of human dental plaques, often forming clusters between the bacterial cells that probably contribute to the biofilm structure [63,64]. Accordingly, the viral particles of the complex bacteriophage community in dental plaques [65] may contribute to biofilm stability and thus be involved in pathogenesis of caries. The phage contribution to biofilm structure, regardless of the mechanism involved, would influence not only the population that produced the viral particles but also other bacterial populations within a multispecies or multi-strain biofilm.

Pf4 and other Pf1-like phage particles can also interact with mammalian macrophages, directing the immune response along an antiviral pathway that is largely ineffective against a bacterial pathogen (reviewed in [66]). Interestingly, the interactions of many other bacteriophages with human or animal immune systems lead to variable outcomes (reviewed in [67,68]), but a detailed analysis of these data is beyond the scope of the present review.

### 2.3. Other POSE Mediated by Phage Particles

Two well-studied phenomena can definitively be attributed as POSE: (1) the modified fate of already infected cells as a result of superinfection such as the lysis inhibition in T-even phages [69,70], (2) the above-described effect of infection multiplicity on the probability of a lysogenic decision in many temperate phages. In the case of the virulent phages which produce special proteins to suppress bacterial defenses, the first infection of a bacterial cell may be, nevertheless, prevented by cellular antiviral systems. However, such an exposure makes the surviving cell transiently more susceptible to reinfection. Such an immunosuppression effect was observed, for example, in phages encoding anti-CRISPR proteins [71].

## 3. POSE Mediated by Quorum-Sensing (QS) Signals

Many phage-host interactions are modulated by bacterial QS systems (recently reviewed in [72]). However, only few cases of QS signals being induced or modified by phage infection have been described. One of the first examples of such interactions was discovered in Sp-beta-like phages of *Bacillus subtilis*. These temperate phages encode their own QS system, referred to as arbitrium. The effectors of this system is a 6 a.a. arbitrium peptide secreted by the infected cells (the sequence of arbitrium is specific to each particular phage). When a sufficient concentration of the peptide accumulates, it is transported into bacterial cells and, in the case of subsequent infection with a phage, stimulates the transition to a lysogenic state [73]. As the result, the phage preferably lyses the cells at the onset of infection of a dense bacterial population, such as a laboratory culture, but then switches to a high frequency of lysogenization. Similar systems were later discovered in a number of other bacteriophages of gram-positive bacteria and in their mobile elements. Interestingly, in some of these phages, such as Wbeta or a Waukesha92-like virus, supplementation by the corresponding arbitrium’s led to prophage induction rather than to the lysogeny stimulation [74]. In one of the Sp-beta-like phages phi3T, a second QS-system, Rapφ-Phrφ, was found. It suppresses the expression of the host’s antiviral systems under the conditions of a low phage density and a high density of sensitive *B. subtilis* cells [75]. The arbitrium and Rapφ-Phrφ systems presumably interact with each other, coordinating not only the lytic/lysogenic decision-taking in different cells, but also gene expression in the resulting lysogenic cells. This promotes cheating behavior, which provides the newly formed lysogens an immediate fitness advantage by delaying the costly production of “public goods” such as biofilm matrix compounds or antimicrobial molecules [75].

In contrast to the specialized effect of the arbitrium system, which was the product of a complex evolutionary development, more general reactions associated with the release of cellular genome products during phage lysis have been described. Thus, during *P. aeruginosa* infection with virulent phages, a modification of the swarmer (social) movement of the bacterial population was documented. Here, the products of phage lysis acted as a repellent. This phenomenon of “collective stress” was caused by the release of the quinolone signaling molecule PQS [76]. A similar effect was observed with antibiotic-induced cell lysis. The avoidance of the phage multiplication site can be interpreted as a defense reaction of the microbial population. Under certain conditions, however, this behavior can have a contrary effect, namely to facilitate the efficient spread of reversibly adsorbed phage and/or infected cells. Also, a viral infection of the local population will clearly stimulate the spread of the bacteria themselves.

General bacterial QS effectors and quorum quenching (QQ) molecules were found to modulate lysis-lysogeny switch in temperate bacteriophages. In most of the cases QS signals enhance prophage induction, though in some phages suppression of lytic growth was observed (recently reviewed in [77]). Therefore, if phage infection can somehow stimulate the release of these compounds, the prophage-depended POSE effects can be expected.

## 4. Phages Modulating the Biofilm Development

A bacteriophage attack may successfully destroy single-species or even multi-species biofilms and many applications of bacteriophages to control biofilms in various environments have been suggested, including the treatment and prevention of biofilm infections in humans or animals (see recent reviews [78,79,80]). Anti-biofilm activity by bacteriophages is believed to base mainly on the direct bactericidal effect of the phage infection combined with the action of phage-encoded enzymes depolymerizing various components of the biofilm matrix. Possible effects of the biofilm dispersal to include the non-infected cells of the biofilm microbial community were considered above. In some phage-host systems, however, more sophisticated interactions were described. Similarly to antibiotics that may stimulate the biofilm growth if applied in sub-inhibitory concentrations [81,82], the increase of biofilm production and stability in the presence of certain bacteriophages has been reported. This is also often associated with limited phage exposure when only a fraction of the cells become infected. 

The interaction of the *Vibrio anguillarum* culture with schizo-T-even bacteriophage KVP40 caused enhanced cell aggregation and stimulated the formation of biofilms, which protected the bacterial population from infection [83]. Once the biofilm is formed, the accumulation of the host QS signal suppresses the expression of the phage receptor, the outer membrane protein OmpK [84]. As a result in the phase of biofilm dispersion, the spreading cells were much less susceptible to infection by the KVP40 phage. Although the mechanism of the latter effect was identified, the nature of the signal that mediated the aggregation of the uninfected cells after bacteriophage addition remains to be elucidated. Interestingly, another *V. anguillarum* phage—the siphovirus ΦH20—markedly suppressed biofilm formation under the same experimental conditions (though in a different host strain). That result highlights that some specific interaction within phage-host systems is responsible for this POSE [84]. 

*E. coli* and *P. aeruginosa* also showed enhanced biofilm formation when treated with low doses of phage, whereas high phage loads decreased the biofilm mass [85]. This indicates that some signal generated by infection of a small fraction of the cells in the culture stimulated the expression of the features (most probably *via* up-regulation of the corresponding genes) necessary for biofilm development that could protect the cells from phage attack. Similar effects have been described for several other host-phage systems [86,87,88]. The mechanisms behind stimulated biofilm formation after phage infection of a part of the population are currently poorly understood. Low MOI infection of *Staphylococcus aureus* by the phage phiIPLA-RODI caused increased biofilm growth and promoted the formation of a more robust biofilm matrix containing more extracellular DNA [89]. Interestingly, this response was associated with an increased expression of certain stringent response genes as determined by the transcriptomic analysis. Nonetheless, it remains unclear whether this response was triggered in the infected or non-infected fraction of the bacterial population. If some of the chemical messengers and/or physical signals originating from the phage infection cause a stringent response in the remaining non-infected cells, this may also slow down the progression of the phage infection in the biofilm.

The effect on cell aggregation and biofilm formation is probably one of the most common types of the POSEs. Multiple molecular and population mechanisms can contribute to such effects in different phage-host systems. The records of biofilm stimulation by low exposure to phages may be considered as an argument for increasing the dosage of the phage preparations if using them therapeutically. Nonetheless, note that complex pharmacokinetics of the phages (reviewed in [90,91]) can preclude the efficient penetration of abundant phages to the infection site or to specific micro-niches colonized by bacteria within this site of the human body. Therefore, the local bacterial sub-populations may experience variable exposure even within one patient. At the same time, not all the phage-host systems show biofilm enhancement even if bacteria are infected with a low initial dose of the phage. Experimentally evaluating the effect of candidate phages on biofilm formation in vitro may be a useful test for selecting the optimal phage strains for therapeutical applications. At the same time, better understanding the mechanisms of this POSE may enable predicting the optimal phage *in silico* on basis of phage genomic sequence.

## 5. Other Possible Mediators of POSEs 

### 5.1. Normal or Phage-Induced Metabolites

A number of different molecules and larger particles that are released during lysis of bacteria or secreted by the infected cells may exhibit physiological activity towards the non-infected cells and mediate specific POSEs.

The release of the molecules normally present in the bacterial cytoplasm can be sensed by surrounding bacteria. On one hand the extracellular ATP can be taken up and used by bacteria as a nutrient. On the other hand it can also mediate communication between microorganisms and the macro-host, and, importantly, between bacteria (reviewed in [92]). The survival of stationary phase *E. coli* and *Salmonella* over 7 days incubation was significantly improved by supplementing the medium with 10μM ATP, that is ca. 100–500 times lower than the intracellular ATP concentration [93]. The extracellular ATP and dATP was shown to stimulate biofilm formation in *E. coli*, *S. aureus*, *Stenotrophomonas maltophila* and *Acinetobacter baumanii* [94]. The same albeit less pronounced effect was reported for dGTP, wth no effect for dCTP or dTTP [93]. In *Fusobacterium nucleatum*, an important periodontal pathogen, extracellular ATP provoked biofilm dispersal and dissemination of the bacteria. In planktonic bacteria communities or cultures, the cells’ cytoplasm makes up only a small fraction of the microcosm volume, and the released ATP will be diluted below active concentration threshold. In contrast, the local concentration of the released ATP may be much higher in structured habitats where the cells are densely packed (e.g., in biofilms or in an infected microcolony in the phage-inoculated bacterial lawn in the laboratory).

Many bacteriophages significantly modify the metabolism of an infected cell, making it a potential producer of unusual compounds [95,96] that can reach and influence non-infected cells. The variety of such compounds includes unusual nitrogenous bases and/or nucleotides that are synthesized during infection by phages with hypermodified DNA [97]. For example, bacteriophage 9 g [98] includes in its DNA large amounts of the unusual base archaeosine, which replaces up to 25% of G residues [99]. Upon the cell lysis free unusual bases non-incorporated in viral nucleic acids are released to the environment and potentially may influence surrounding cells. Interestingly, the archeosine analog queosine and their common precursor PreQ0 were shown to be physiologically active at least in eukaryotic systems, where they potentially act as antioxidative compounds [100,101]. At the same time, being a virulent phage, bacteriophage 9 g very efficiently forms highly stable associations with different host strains, referred to as carrier state cultures or pseudolysogenic associations (PA) [98]. All the cultures obtained by transferring the material from the plaques of 9 g phage retain the ability to produce phages over a large (probably indefinite) number of the passages on solid media. In contrast, in most of the other tested phage-host systems, only a fraction of such cultures is stable and only for a limited number of the passages ([102] and our unpublished data). It is currently unknown whether there is any connection between these unusual properties of this virus and the type of DNA modification present in it. Nonetheless, it is logical to assume [103] that some product of phage infection diffusing from the center of the growing plaque can stimulate the formation of cells phenotypically resistant to phage, or cells with phenotypically reduced sensitivity to phage (for example, expressing fewer phage receptors). This can provide a basis for PA stabilization and serve as an intermediate for subsequent selection of resistant genotypes of bacteria.

### 5.2. Peptides

A vast evidence of peptides exhibiting physiological activity towards bacteria has been accumulated. This includes antimicrobial peptides and peptide antibiotics [104,105], small bacteriocins [106,107], membrane-penetrating peptides [108], anti-biofilm peptides [109] and others. Many peptides effectively penetrate the bacterial cell wall, exerting their influence after having been supplied externally. Little, however, is known about the possible physiological activity of the peptides formed due to protein degradation and released upon bacterial lysis. The sequences of the peptides with known activity often closely match fragments of bacteria or bacteriophage sequences. This leads to the assumption that the bacterial cell lysis may exhibit certain activity towards bacteria of the same or of other species. Bacteriophages, especially large virulent viruses, encode a large number of small ORFs, whereby the function of the majority remains unknown. Some such proteins and peptides participate in host macromolecular synthesis shutdown [110,111]. To my knowledge, no POSEs mediated by the peptides generated by phage infection (except for peptide QS effectors, some of which were mentioned above) were described so far. Nonetheless, a systematic search for such effects may well reveal them. 

### 5.3. Small RNAs

Many studies have demonstrated the emergence of cells acquiring partial and/or transitory resistance (tolerance) to phages [83,103,112,113,114]. Such clones have often been observed under conditions of relatively low phage infection pressure, e.g., infection initiated at low MOI, and arise with a greater frequency in stationary phase cultures. Lacqua et al. [115] associated the formation of *E. coli* cells tolerant to two unidentified bacteriophages with increased expression of curli, which caused cell aggregation. At the same time, this phenotype was associated with overproduction of the Dps protein. Dps is a nucleoide-associated protein involved in DNA compaction and is expressed mainly during the cell transition to the stationary phase or as a stress response [116]. Nevertheless, in the study [115], large amounts of Dps were recorded in the outer membrane preparations, which was also accompanied by the disappearance of the major outer membrane protein OmpX. Moreover, the host *dps* mutants did not produce phage-tolerant clones. The decrease of OmpX may explain the phage tolerance if this protein or some other protein(s) or structure(s) associated with it, is involved in the reception of the phage. However, the role of the Dps protein is unclear. Interestingly, in a recent study [117], Dps protein was shown to have RNA-binding activity. Taken together, these results enable formulating the hypothesis that, in the phage-host system studied by Lacqua et al. [115], the RNA is involved in alerting some as yet non-infected host cells to induce a phage-tolerant phenotype. 

In recent years, data have been accumulating on the significant role of bacterial small noncoding RNAs in regulating cell metabolism [118], and, interestingly, in the interactions between symbiotic bacteria and their plant and animal macroorganism hosts (see reviews [119,120]). Regulatory microRNAs also have been found in bacteriophages, where they play an essential role in controlling phage gene expression and, often, in making lytic or lysogenic decisions [121,122]. Until recently, it was believed that bacteria do not secrete small RNAs. More recent works by the Ozoline group, however, showed that *E. coli*, as well as other bacteria, can specifically secrete small RNAs, and that this process is specific in relation to well-defined types of these molecules (although the mechanism remains unknown). The spectrum of secreted RNAs depended on the presence of other microorganisms in the culture [123]. Moreover, some small RNAs and their synthetic analogs were able to effectively penetrate bacterial cells, exerting a physiological effect on them. For example, RNA secreted by *Rhodospirillum rubrum* and *Prevotella copri* during co-cultivation with *E. coli* was able to suppress the growth of the latter [124]. Taking into account the data of Lacqua [115], this suggests that the Dps protein is involved in importing small RNAs into the cell (hypothesis suggested by O. Ozoline, IBFM RAS, personal communication).

Note here that, during phage lysis, the entire transcriptome of an infected cell enters the external environment. Considering the above-cited data on the physiological activity of several cellular RNAs, it is very likely that other RNA-mediated POSE may also be present. Interestingly, some bacteriophages, during their lytic development, intensively express certain RNAs that have no obvious function. For example, the vB-SaU-515F1 staphylophage transcriptome [125] contains a very high amount of transcripts from one of the intergenic regions of the phage genome that has no known physiological function.

### 5.4. Extracellular Membrane Vesicles

The transfer of small RNAs and other active compounds can be mediated by the membrane extracellular vesicles (EV) released from the outer membrane by Gram-negative bacteria or from the cytoplasmic membrane by Gram-positive species [126]. The EV frequently contain small RNAs and can deliver them to eukaryotic and, possibly, also to bacterial cells [127,128,129]. The ability of the EV to fuse to the membranes of the other bacterial cells may also modify the recipient cells surface. The co-culturing of the *B. subtilis* strain sensitive to the SPP1 phage with the resistant strain lacking the SPP1 receptor led to the acquisition of transient phage sensitivity. This was due to trafficking of the receptor *via* the cytoplasmic membrane vesicles released from the sensitive cells [130].

EV production is a normal physiological process in many bacterial species. Although the mechanisms of EV production by various budding processes not associated with the death of EV-producing cells were described, in some cases lysis-associated EV production is also present (recently reviewed in [126]). The latter process (leading to formation of the functional EV) frequently involves activation of prophage-related lysis systems [126,131,132,133]. The formation of the physiologically active EV is therefore highly likely in the case of external infection by bacteriophages. Interestingly, the membrane vesicles formed during the lysis of Gram-negative bacteria by many phages are markedly different in composition from the EV produced by budding from the OM surface. Most such phages, in addition to the holin-endolysin lysis system, encode spanins. These proteins link the outer and inner membranes of the infected cell and, after peptidoglycan removal by endolysin action, induce membrane fusion to form hybrid vesicles (reviewed in [134]). The vesicle formation is especially effective in case of phages harboring SAR-type endolysins that degrade peptidoglycan all over the surface of the lysing cell [134].

Importantly, the EV have many different functions in bacterial physiology including communication within bacterial populations, communication between bacteria and higher organisms, components of biofilm matrix, and nutrient acquisition. etc. (for a review see [126,135]). Therefore, the POSE associated with extensive EV formation during phage lysis should be present, although they have apparently not yet been demonstrated experimentally. Finally, a large fraction of bacterial QS signals that may be represented by lipophilic molecules are in fact transported by the EV [132,136], see also review [137]. Since the QS is known to be involved in many phage-host interactions (recently reviewed in [72]), the increased signal trafficking should impact the reaction of the microbial community to the ongoing phage infection.

## 6. Influence of Phage Selection on the Genetic Heterogeneity of Resistant Populations

The genetics of the coevolution of bacteria and bacteriophages in various experimental and natural systems has been studied in detail over the last two decades. The present review restricts itself to considering the possible effects of high-intensity selection, corresponding to the phage therapy situation, on the diversity of surviving bacteria. This effect of a phage infection may be considered as a variant of the POSEs because the part of the population initially resistant to the phage is not exposed to infection, but the elimination of susceptible bacteria changes its genetic landscape. This review does not consider the pleiotropic effects of mutations that provide the resistance (on which there is a fairly extensive literature; see [29] for review), but focuses instead on the study of genetic changes in bacteria not directly related to phage resistance.

The classic work of Luria and Delbruck [138] showed that mutations in *E. coli* conferring the resistance to T1 phage arise spontaneously before the bacterial population was brought in contact with the virus (on the phage agar). Under the conditions of that experiment, the probability of selection of resistant mutants through intermediate stages in the course of the local microevolutionary process was minimized. Apparently, when using a “highly virulent” phage in high concentration, only those cells survived that were completely resistant at the moment of stringent selection performed on the plates supplemented by high dosage of phage. However, under a milder phage selection pressure in natural habitats, different adaptation trajectories are possible. 

Coevolution of the *P. fluorescens*–SBW25 phage system in a laboratory microcosm led to the selection of hypermutable host variants [139], which had a significant advantage over the wild-type phage [140]. At the same time, under near-natural conditions no selection of hypermutable variants occurred [141]. Probably the negative effect of genetic instability such as accumulation of deleterious mutations outweighed the advantages of better adaptability to the phage. In these studies only the occurrence of stable hypermutable phenotypes resulting from mutations of the proteins responsible for genome stability (such as MutS) was analyzed. However, in some microbial species or strains the hypermutability can be a transient state rather a phenotype linked to specific stable genetic alteration(s). For example, in some *Streptococcus pyogenes* strains the mutator phenotype is controlled by reversible integration-excision of a prophage-like element into the *mutS* gene [142]. The hypermutable phenotype can also emerge in response to different kinds of environmental stress [143], and, therefore, is likely to be induced spontaneously due to the noise of regulation circuits. As it was discussed above, some hypothetical POSEs can be linked to the stress response pathways activation thus contributing to bacterial genetic diversity by inducing transient hypermutable phenotype.

Noteworthy, even without any active induction of mutator phenotypes, intensive phage selection may increase the fraction of the genetic lineages with recent history of transient hypermutability. No literature data are available on the selection of hypermutable variants in the case of “one-hit” selection such as by plating on phage agar. Our unpublished data, however, indicate that a significant portion of the phage-resistant mutants often carry large numbers of polymorphisms that are not obviously related to the phage-host interactions and therefore could have passed through the hypermutability phase. This suggests that, in culture, phenotypically hypermutable cells may exist, whereas their progeny have a normal mutation frequency. If this assumption is correct, then such a mechanism could enable bacteria to bypass the negative effect of the perpetual hypermutable phenotype.

## 7. Concluding Remarks

Here we described multiple examples where microbial populations respond to phage infection by modifying the physiology of uninfected bacterial cells. It is not yet clear whether these examples are exotic exceptional cases or are manifestations of more general overlooked principles. 

The development of new screening tools enabling the differential analysis of physiological alterations of adjoining, non-infected cells would help to shed the light on this question. The pre-setting of the indicators of a general “physiological state” of bacterial cells is challenging. These may involve simple characteristics such as cell morphology and growth rates. The reporter systems indicating the activation of the known general stress pathways—such as stringent response, stationary phase gene expression, SOS-response, oxidative stress response, surface stress and heat-shock response—are relatively simple to develop. Systematically identifying the POSE and deciphering the underlying molecular mechanisms will improve our understanding of the principles behind the shaping of microbial communities by bacteriophage infection. This would be an important step forward in successfully harnessing bacteriophages to combat non-desirable bacterial populations, including phage therapy, food spoilage prevention and other phage-based technologies.

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
