# Peer review of "The Burden of Survivors: How Can Phage Infection Impact Non-Infected Bacteria?"

_ijms, 2023, doi:10.3390/ijms24032733_

Round 1

Reviewer 1 Report

In this review authors have explored how different agents causing POSE (physiology of survivors effects) can affect the interactions between bacteriophage and bacteria which are not directly infected. These agents can vary ranging from phage particles or phage structural proteins, quorum sensing (QS) signals, metabolites, peptides, small RNAs and extracellular membrane vesicles. Overall, the review is well written. My suggestion is to carefully go through the manuscript and make following textual changes:

1.  Line 41-42 “when the focus of the attention microbiologists”. Please rephrase as it sounds incorrect.

2. Line 118 correct spelling of “particals”.

3. Line 125 change “in similar” to “is similar”.

4. Line 133 correct spelling of “scpecial".

5. Line 141 correct spelling of “specieas”.

6. Line 191 change “are modulated the bacterial” to “are modulated by the bacterial”

Author Response

Please find below point-by-point responses to the Reviewer1 comments

In this review authors have explored how different agents causing POSE (physiology of survivors effects) can affect the interactions between bacteriophage and bacteria which are not directly infected. These agents can vary ranging from phage particles or phage structural proteins, quorum sensing (QS) signals, metabolites, peptides, small RNAs and extracellular membrane vesicles. Overall, the review is well written. My suggestion is to carefully go through the manuscript and make following textual changes:

We’re grateful to the Reviewer for the positive evaluation of our work and for spotting the typos. We also performed a round of deep editing of the language and style over the whole MS with the help of a native English-speaking professional editor.

  1.  Line 41-42 “when the focus of the attention microbiologists”. Please rephrase as it sounds incorrect.

The sentence was changed as follows: Interestingly, the bacteriophage discovery took place during an era when the conceptualization of a microorganism by microbiologists was shifting from bacterial culture to bacterial cell as a representation of an individual organism [10].

  1. Line 118 correct spelling of “particals”.

Done

  1. Line 125 change “in similar” to “is similar”.

Done

  1. Line 133 correct spelling of “scpecial".

Done

  1. Line 141 correct spelling of “specieas”.

Done

  1. Line 191 change “are modulated the bacterial” to “are modulated by the bacterial”

Done

Reviewer 2 Report

The topic of this paper is interesting and significant and I was eager to read to the paper. The majority of the ideas the authors are trying to convey in the paper however are lost in the sentence and paragraph structures. I recommend working with someone to improve sentence and paragraph structure to more clearly communicate the authors' ideas.

Some example suggestions for edits:

Introduction: I recommend re-writing introduction to improve sentence structure and clarity of the ideas you want to convey.

Lines 28 – 30: Remove the close parentheses after “Twort [1]”

Lines 28 – 30: this sentence has multiple grammatical problems and I am also not entirely sure what the authors are trying to communicate so I couldn’t take a stab at correcting it. “…served as a valuable model”? “that allowed to established a concept” doesn’t work. What does it mean to be a “transmissive genetic program”?

The bacteriophages discovered more than 100 years ago by F. Twort and F.

D’Herelle have served a valuable model that allowed to established a concept of the

virus as a transmissive genetic program (reviewed in [3]).

Line 32: what “phenomena” are you referring to? The idea in this sentence is not coming through clearly.

Lines 36 – 38: This sentence has too many verbs in it and is unclear. Phage and bacterial relationships in nature are no longer considered solely host-parasite or predator-prey but rather multifaceted interactions that include ….etc

Line 39: I don’t know what the phrase “From the perspective of contemporary knowledge” means.

Line 41-43: re-write for clarity.

Chapter 2

Line 85:  what is PT?

Lines 79-81 – rewrite for clarity. Plaques do not form around bacteriophages.

Lines 85-87: How was the T7 phage engineered? Is that relevant to the activity of DspB? Was it engineered to express DspB?

Lines 91-92: Find ways to re-write sentences that simplify sentence structure and improve clarity. Example: Phage encoded enzymes that degrade bacterial capsules and lack direct bactericidal effects may have in vivo therapeutical effects by exposing bacteria to immune factors.

Lines 91-92: This sentence about exposure of cells to immune factors doesn’t serve as a very strong topic sentence for this paragraph. The paragraph focuses on the degradation of biofilm and its impact on cell signaling in the microbial population. Two topics in one paragraph.

Lines 103-105: re-write for clarity.

Lines 105-106: re-write for clarity.

Lines 103-112: I think this paragraph is about the development of phage defense escape mutants? Are Pf4 phage mutants developing the ability to superinfect Pf4 lysogens or are non-Pf4 phage mutants overcoming repressor-independent defenses and superinfecting?

Lines 115-117: rewrite for clarity. It’s possible that lytically or lysogenically infected cells impact the lysogenic decision of adjacent cells through the release of diffusible signals.

Lines 119-121: run on sentence. The frequency of lysogeny increases in cells infected by multiple lambdoid phages. This is likely explained by higher gene dosage of CII, the transcription factor that activates the lysogenic pathway.

Lines 124-125: A similar mechanism may be expected in many other phages in which lysogeny control

is similar to the phage lambda.

Lines 129-132: This is a single sentence paragraph. It refers to the mechanism of pattern formation. Is it referring to the pattern formation of plaques in the previous paragraph?  I’m lost.

Lines 133-134: Three spelling errors: “special” “sophisticated” and “opportunity.”

Lines 133-148: This paragraph seems to have multiple topics or at least needs a better topic sentence so that the reader has a better sense of how to digest the content of the paragraph. Is this paragraph about phage killing of sensitive cells to allow lysogens to fill niches or is it about how phages penetrate dense microbial populations and reach sensitive hosts via transport by non-sensitive microbes including bacterial and fungal species?

Chapter 2.2

Lines 151 and 152: This is a very specific sentence to start this paragraph. Orient the reader to what this paragraph is about. Also specify whether Pb1A and B are phage or host-encoded (presumably these are host encoded).

Author Response

Please find below point-by-point responses to the Reviewer2 comments

The topic of this paper is interesting and significant and I was eager to read to the paper. The majority of the ideas the authors are trying to convey in the paper however are lost in the sentence and paragraph structures. I recommend working with someone to improve sentence and paragraph structure to more clearly communicate the authors' ideas.

We are grateful for the Reviewer for his/her high evaluation of the concept of our review. We do not fully understand the concern of the Reviewer on that “majority of the ideas the authors are trying to convey in the paper however are lost in the sentence and paragraph structures”. From our point of view and from the point of view of colleagues who helped us to prepare the manuscript (namely Prof. M.R.J. Clokie and Dr. E.E. Kulikov mentioned in the Acknowledgements) the text is quite straightforward and easy to understand. However, we addressed all the points the Reviewer mentioned and also tried to improve overall style of the MS. We also performed a round of deep editing of the language and style over the whole MS with the help of a native English-speaking professional editor.

Some example suggestions for edits:

Introduction: I recommend re-writing introduction to improve sentence structure and clarity of the ideas you want to convey.

The style was revised all over the MS

Lines 28 – 30: Remove the close parentheses after “Twort [1]”

Done

Lines 28 – 30: this sentence has multiple grammatical problems and I am also not entirely sure what the authors are trying to communicate so I couldn’t take a stab at correcting it. “…served as a valuable model”? “that allowed to established a concept” doesn’t work. What does it mean to be a “transmissive genetic program”?

The bacteriophages discovered more than 100 years ago by F. Twort and F.

D’Herelle have served a valuable model that allowed to established a concept of the

virus as a transmissive genetic program (reviewed in [3]).

The fragments was re-written as follows:
The bacteriophages discovered more than 100 years ago by F. Twort [1] and F. D’Herelle [2] turned to be a valuable model for investigating the nature of viruses and studying the basic molecular mechanisms of life. Bacteriophage research was the main approach that eventually led to the currently established concept of the virus as a transmissive genetic program (reviewed in [3]).

We would like to note that the hypothesis of the virus as a “transmissive gene” is a historical formula appeared in 1920s. We slightly modified it to “transmissive genetic program” to adapt it to the contemporary definition of a gene.

Line 32: what “phenomena” are you referring to? The idea in this sentence is not coming through clearly.

We meant all the related biological phenomena that take place in the living world. For clarity we modified the phrase as follows:

Today, bacteriophage biology has developed into one of the most elaborate fields of virology, which is able to establish the link between the phenomena observed at all levels of the biological organization ranging from the sub-molecular to the biosphere.

Lines 36 – 38: This sentence has too many verbs in it and is unclear. Phage and bacterial relationships in nature are no longer considered solely host-parasite or predator-prey but rather multifaceted interactions that include ….etc

In fact, this composite sentence had only two verbs (or verb constructions), namely “now being replaced” and “often may be considered”. To improve the readability we have split it into two sentence as follows:

Our understanding of phage and bacteria relationships in nature as being solely host-parasite or predator-prey connections is being replaced by the concept of multifaceted interactions. These interactions often may be considered as mutualistic according to their outcomes for both bacterial and viral populations [7-9].

Line 39: I don’t know what the phrase “From the perspective of contemporary knowledge” means.

We have in our view the idea that an enlighten intellectual, who masters well the contemporary bacteriophage biology, may call the phages “natural frenemies”  instead of  “natural enemies” of bacteria. However, in order to avoid involvement of this imagined informed person, we used the passive voice and, from our point of view, the idea is clear. However, we slightly changed the wording for clarity

Now this phrase stays as follows:

Based on the perspective of contemporary knowledge, bacteriophages may therefore be called “natural frenemies” rather than “natural enemies” of bacteria.

Line 41-43: re-write for clarity.

This fragment was modified as follows:
Interestingly, the bacteriophage discovery took place during an era when the conceptualization of a microorganism by microbiologists was shifting from bacterial culture to bacterial cell as a representation of an individual organism [10]. 

Chapter 2

Line 85:  what is PT?

Lines 79-81 – rewrite for clarity. Plaques do not form around bacteriophages.

PT stands for phage therapy. We apologize for the fact that the line introducing this abbreviation upstream was deleted during the editing.

We meant the formation of the halo aroung the plaques of bacteriophages. The whole fragment was modified as follows
The formation of a halo around plaques of some bacteriophages was one of the first described POSE mediated by bacteriophage structural proteins. The halo-forming phages encode polysaccharide-degrading enzymes which help them to penetrate the capsules and/or exopolysaccharides produced by their hosts. Most of the known phage polysaccharide depolymerases are phage structural proteins such as tail spikes. During the infection these proteins may be produced in significant excess, spreading out of the viral infection site (e.g. the plaque) faster than phage particles. This modifies the surface of the cells in the non-infected area of the bacterial lawn, adjacent to the plaques.

Lines 85-87: How was the T7 phage engineered? Is that relevant to the activity of DspB? Was it engineered to express DspB?

The technical detail of the modification falls out of the scope of our review. The dspB gene was introduced into the phage genome to allow expression of soluble enzyme (not connected to the virions) in the infected cells.

We modified the sentence as follows:

The engineered T7 phage that produced in the infected cells the soluble enzyme DspB degrading the E. coli cell-bound exopolysaccharide adhesin (polymeric β-1,6-N-acetyl-d-glucosamine) had ca. 100-fold stronger anti-biofilm activity compared to the non-modified virus [33].

Lines 91-92: Find ways to re-write sentences that simplify sentence structure and improve clarity. Example: Phage encoded enzymes that degrade bacterial capsules and lack direct bactericidal effects may have in vivo therapeutical effects by exposing bacteria to immune factors.

Lines 91-92: This sentence about exposure of cells to immune factors doesn’t serve as a very strong topic sentence for this paragraph. The paragraph focuses on the degradation of biofilm and its impact on cell signaling in the microbial population. Two topics in one paragraph.

The whole fragment was modified

Removing the capsules in the context of natural environments may alter interactions of the bacteria to various external factors. For example, capsule-degrading enzymes were shown to lift the protection of microbial cells against immunity factors mediating the in vivo therapeutical effect of phage-derived tail spikes which lack any direct bactericidal activity in vitro [30, 31]. The enzymes produced by the phage-infected cells in the course of phage therapy (PT) may thus contribute to the elimination of the non-infected pathogen cells by the immune system.

The disruption of the biofilm structure by bacteriophage-encoded enzymes [32] may also contribute to PT efficiency as well as cause POSEs in natural ecosystems. The engineered T7 phage that produced in the infected cells the soluble enzyme DspB degrading the E. coli cell-bound exopolysaccharide adhesin (polymeric β-1,6-N-acetyl-d-glucosamine) had ca. 100-fold stronger anti-biofilm activity compared to the non-modified virus [33]. Clearly, degrading the matrix of a multi-species biofilm based on the phage infection of only one of microbial species may have strong and multifaceted impact on the co-habitant species that are not sensitive to this phage. The physiological state of bacteria in a biofilm is often different from the physiological pattern of planktonic cells (see [34, 35] for reviews).

Lines 103-105: re-write for clarity.

Changed as follows
Biofilm disruption due to the action of phage-encoded enzymes is merely one specific example of a class of POSE relying on a modification of spatial structure of microbial communities.

Lines 105-106: re-write for clarity.

The fragment was modified.

Biofilm disruption due to the action of phage-encoded enzymes is merely one specific example of a class of POSE relying on a modification of spatial structure of microbial communities. Temperate filamentous Pf4-like bacteriophages of Pseudomonas aeruginosa are involved in various processes during the biofilm development by their host, including the formation of the correct biofilm spatial organization. During the initial stages of biofilm development by the PAO1 strain lysogenic for Pf4, the so-called superinfectious variants (mutants) of the phage emerge in the growing microcolonies settled on the surface colonized by bacteria. These superinfectious variants are capable of infecting cells, lysogenic by Pf4. Nonetheless, they act in a selective manner, mediating the death of some cells in the middle of the forming microcolonies but leaving the cells at the microcolony periphery uninfected. Such selectivity is based on different expression of the pili serving the Pf4 receptors. The spatially controlled death of a part of the microbial population yields biofilm channels [41, 42]. The surviving bacterial cells, and possibly other cells in multi-species biofilms, benefit from the more robust biofilm following the optimal spatial design [41].

Lines 103-112: I think this paragraph is about the development of phage defense escape mutants? Are Pf4 phage mutants developing the ability to superinfect Pf4 lysogens or are non-Pf4 phage mutants overcoming repressor-independent defenses and superinfecting?

Up to our knowledge, the biology of the superinfectious variant of Pf4 is not yet completely understood. There was a publication linking the ability to infect lysogens to the repressor mutation. However, it was demonstrated that these variant reproducibly emerge during the biofilm establishment and, while super-infecting the cells already carrying Pf4 prophage cause their death, in contrast to Pf4 infection of naïve cells, in which the filamentous phage multiplies without killing the host.
However, the in-depth consideration of the molecular basis of Pf4 superinfectious phenotype falls out the scope of our review.

Lines 115-117: rewrite for clarity. It’s possible that lytically or lysogenically infected cells impact the lysogenic decision of adjacent cells through the release of diffusible signals.

The fragment was modified as follows
The mechanism of the concentric ring formation is most probably based on a diffusible signal(s) produced by the cells that undergo lytic or lysogenic infection. These signals are sensed by other cells to alter the probability of lysogenization upon contact with the phage.

Lines 119-121: run on sentence. The frequency of lysogeny increases in cells infected by multiple lambdoid phages. This is likely explained by higher gene dosage of CII, the transcription factor that activates the lysogenic pathway.

The fragment is modified as follows:
The probability of the lysogenic decision increases substantially in those cells simultaneously infected by multiple particles of the lambdoid phages [46-48]. The likely explanation is a higher gene dosage of CII, the key transcription factor activating the lysogenic pathway.

Lines 124-125: A similar mechanism may be expected in many other phages in which lysogeny control

is similar to the phage lambda.

Corrected

Lines 129-132: This is a single sentence paragraph. It refers to the mechanism of pattern formation. Is it referring to the pattern formation of plaques in the previous paragraph?  I’m lost.

Exactly! The discussion of the above-mentioned story of the patterns in the temperate phage plaques continues. The phrase was modified as follows:
Regardless of the mechanism behind the concentric pattern formation, if phage-insensitive microorganisms are present in the system, then such a pattern should create for them the alternating zones of higher and lower intensity of the competition with the lysogens formed from the phage-sensitive strain. (This assumes that the density of the sensitive cells escaping from infection and/or phage-resistant mutants is much lower.)

Lines 133-134: Three spelling errors: “special” “sophisticated” and “opportunity.”

Corrected

Lines 133-148: This paragraph seems to have multiple topics or at least needs a better topic sentence so that the reader has a better sense of how to digest the content of the paragraph. Is this paragraph about phage killing of sensitive cells to allow lysogens to fill niches or is it about how phages penetrate dense microbial populations and reach sensitive hosts via transport by non-sensitive microbes including bacterial and fungal species?

We make a parallel between two effects: (i) ability of the lysogens come to a site populated by sensitive bacteria, to kill a part of them due to the infection by the phage, released by the invading population and to settle up taking this window of opportunity and (ii) the ability of some bacteria to get loaded by phages and to carry them to another place where released phages may also cause the lysis of the sensitive resident population and to facilitate the invasion of the carriers. The ability of some of bacteria to traffic phages along the hyphae strikes us as an amazing and relevant fact.
We slightly modified the fragment to improve its readability

Both mathematical and experimental modelling have shown the increased ability of the lysogenic bacteria to invade the environmental niches densely populated by those bacteria sensitive to the corresponding phage [50, 51]. Although this phenomenon is beyond the scope of the present review (because the lysogens that release the phages were formed as a result of direct infection ), quite a similar bona fide POSE mechanism was recently described for some motile bacteria (Pseudomonas putida, Viridibacillus sp., Enterobacter sp., Serratia sp., Bacillus sp., Janthinobacterium sp.) that are able to non-specifically adsorb bacteriophages infective for other species, such as E. coli [52]. Such phage-loaded bacteria may transport viruses over considerable distances. Interestingly, some bacterial species were found to be able to travel and to transport phages along the hyphae of fungi, increasing their spread in viscosous media such as top agar in the laboratory model. The phages liberated from the carrier cells’ surface may cause local lysis of the resident population (e.g. formation of a plaque on the sensitive strain lawn). This makes room for colony formation by the invading species. Note, however, that the same bacteria, even when assisted by the fungal mycelium, were unable to establish colonies in the absence of the phage: they were outcompeted by the dense and rapidly growing background population (E. coli lawn). This novel trait of an increased ability for invasion aquired by bacteria insensitive to the bacteriophage (as a result of the phage infection of different bacterial species) exactly matches our definition of POSE.

Chapter 2.2

Lines 151 and 152: This is a very specific sentence to start this paragraph. Orient the reader to what this paragraph is about. Also specify whether Pb1A and B are phage or host-encoded (presumably these are host encoded).

These are phage-encoded proteins that can be released from the cells lysing due to prophage induction or lytic infection by the phage.
We added an ‘orienting’ sentence at the beginning of the section:
The bacteriophage proteins released from infected cells may bind to the surface of non-infected cells (or non-induced lysogens) of the same or different species and perform specific functions similarly to the own bacterial proteins delivered to the cell surface from the cytoplasm.

…and one more in the middle of the paragraph:

As discussed above, the whole phage virions also may bind bacterial cells without infecting them. Such surface-exposed virions potentially convey some new features to bacteria, for example by serving as adhesins. Currently, however, we know of no such examples in which whole-virions function as a decoration of bacterial cells, the involvement of some phages into building of such complex structure as the extracellular matrix of microbial biofilms. 

Round 2

Reviewer 2 Report

This paper describes an important and likely under explored topic of how bacteria respond at a population level to phage infection. The review covers various ways phage infection or phage products impact the physiology of non-infected members of a bacterial population and highlights areas that should be more thoroughly investigated. A better understanding how bacteria respond at the population level to phage infection could improve phage applications relevant to human health such as phage therapy.

Introduction

·      Line26-27: change “turned to be” is a strange phrase. Change to “became a valuable model…”

·      Line 55: Change “has led” to “led”

·      Line 63: the word “and” twice in a row. Delete one.

·      Line 64: maintain same tense. “in such a way as to encourage mass killing of the target bacterium and prevent viruses from showing their natural inclination to “make a deal”…A more succinct sentence might be “requires application strategies that allow mass killing of the target bacterium while preventing viruses from …”

·      Line 64: what does it mean to make a deal with the bacterial population? Are you referring to the development of resistance to lytic infection? Make this more clear.

·      Lines 67-70. This is a single sentence paragraph. Try to tie this into the next paragraph where you are trying to communicate the lack of research on uninfected cells within a population of bacteria undergoing viral infection.

Chapter 2:

·      Line 91: Change to “the formation of halo around plaques formed by some bacteriophages”

·      Lines 104-109: These two sentences should be re-written for clarity. A T7 phage was engineered to express DspB in infected cells, which degrades cell-bound exopolysaccharide adhesin. Although the virus targets a single species, this resulted in 100-fold stronger antibiofilm activity in multi-species biofilms relative to non-modified virus.

·      Line 152-156. Why do these two sentences constitute a paragraph? It seems like it belongs with the topic in the following paragraph.

·      Line 178-180: Re-write this sentence for clarity. I lost the plot starting at “similarly to the own bacterial proteins delivered to the cell surface from the cytoplasm.” Do you mean they perform specific functions that mimic the function of bacterial cell surface proteins?

·      Lines 191-193: run on sentence. Re-write for clarity.

Chapter 3

·      Line 234: Change to “is a 6 a.a. arbitrium peptide…”

·      Line 235: Change to “When a sufficient concentration of the peptide accumulates…”

Chapter 4

·      Line 260: Why is this chapter 4 while the above and below sections are labeled as chapter 3?

Chapter 3 (the second chapter 3) – Should this be chapter 5?

·      Renumber all of the sub-sections. The current order is 3.1, 3.3, 3.2, 3.3 (again).

·      Line 322: change to “with 10 mM ATP…”

·      Line 329: corrects spelling of “threshold”

·      Line 332-350: It is not clear how phage-specific molecules such as base analogs, effect non-infected cells in the population. Make this idea more obvious.

·      Line 376: define OM or spell out “outer membrane.”

·      Lines 368-381: connect the dots more clearly for the reader. Phage-tolerant E. coli cells associated with high Dps à Dps accumulates in outer membrane and there is loss of OmpX à mutants of Dps do not become phage-tolerant à Dps has RNA binding activity à how does this take you to the hypothesis in your last sentence?

·      Line 322: this sentence is awkward due to the phrase “which is especially interesting.” I had to read a few times to unwrap it. Suggestion: In recent years, data has accumulated on the significant role that bacterial non-coding RNAs play in regulating cell metabolism and, interestingly, in the interactions between symbiotic bacteria and their plant and animal macroorganism hosts.

·      Line 330: italics for E. coli

·      Line 342: change to “The transfer of small RNAs and other active compunds…”

·      Line 351: typo in the word “also” (alos).

·      Line 351: Do you mean “latter” here rather than “later?”

·      Line 358-359: incomplete sentence. This process is what or especially what?

·      Line 360-361: Delete phrase “It has to be highlighted” and simply highlight the idea by telling us the idea. Also remove inappropriate preposition “into.” Suggestion to reduce wordiness and odd verb tenses: The EV has many different functions in bacterial physiology including communication within bacterial populations, communication between bacteria and higher organisms, components of biofilm matrix, and nutrient acquisition.

·      Line 375: change preposition to “on which there is fairly extensive literature”

·      Line 382: do you mean “at the moment of selection”? Otherwise it sounds like the cells are resistant to the moment of selection.

·      Line 384-385: what do you mean when you say that hypermutable host variants had a significant advantage over the wild type phage? The ideas in this two-sentence paragraph (which may be too short to communicate a significant idea) are not clearly communicated.  In general, the first sentence should communicate the big idea of the paragraph followed by at least 2-3 sentences that provide evidence supporting that big idea. What is the most important idea about the outcome of low MOI infections that you want to communicate and how can you more specifically illustrate this idea with evidence. In your last sentence of this paragraph you state that “under conditions close to natural conditions, the negative effect of genetic instability seems to outweigh the advantages of better adaptability to the phage….” What are the negative effects of genetic instability (deleterious mutations?) and how do they outweigh advantages? These sentences need more specificity in order to clearly communicate your idea. I don’t really know a lot about this topic but presumably the idea here is that the stress of phage infection results in a hypermutable state of the bacterial host (either due to the phage-encoded recombinases or simply due to stress of infection itself) that lead to mutations provide resistance to phage infection???

·      Lines 388-393: Is this the same topic as the previous paragraph (hypermutable host variants) and can this be combined with above paragraph to form a clearer communication of the development of resistance to lytic infection due to hypermutable community members?

Conclusions

·      Lines 396-398: awkward and wordy sentence masks the big idea of this sentence. Re-write to clarify. Suggestion: Here we described multiple examples where microbial populations respond to phage infection by modifying the physiology of uninfected bacterial cells. It is not yet clear whether these examples are exotic exceptional cases or are manifestations of more general overlooked principles.

·      Lines 398-399: The development of what kind of screening tools? Fluorescent time lapse microscopy for example?

Author Response

We would like to express our deep gratitude for extensive efforts made by the Reviewer to help to improve the quality of the manuscript. We accepted all the technical corrections suggested by him/her and tried to elaborate the fragments which appeared to the reviewer not clear enough.

Please find point by point answers to the Reviewer’s comments below

Comments and Suggestions for Authors

            This paper describes an important and likely under explored topic of how bacteria respond at a population level to phage infection. The review covers various ways phage infection or phage products impact the physiology of non-infected members of a bacterial population and highlights areas that should be more thoroughly investigated. A better understanding how bacteria respond at the population level to phage infection could improve phage applications relevant to human health such as phage therapy.

Introduction

  • Line26-27: change “turned to be” is a strange phrase. Change to “became a valuable model…”

Done

  • Line 55: Change “has led” to “led”

Done

  • Line 63: the word “and” twice in a row. Delete one.

Done

  • Line 64: maintain same tense. “in such a way as to encourage mass killing of the target bacterium and prevent viruses from showing their natural inclination to “make a deal”…A more succinct sentence might be “requires application strategies that allow mass killing of the target bacterium while preventing viruses from …”

Changed for the suggested more succinct phrase

  • Line 64: what does it mean to make a deal with the bacterial population? Are you referring to the development of resistance to lytic infection? Make this more clear.

This is explained in the line above “In natural habitats, however, bacteriophages tend to establish a long-term co-existence with sensitive bacterial hosts [7,26,27] rather than driving the host population to extinction.” We meant pletora of strategies of co-existence, not all of them are mechanistically clear, e.g. well known carrier state phenomenon remains poorly understood for many phage-host systems. We do not think the vast detalization of this point would be appropriate here.

  • Lines 67-70. This is a single sentence paragraph. Try to tie this into the next paragraph where you are trying to communicate the lack of research on uninfected cells within a population of bacteria undergoing viral infection.

Done

Chapter 2:

  • Line 91: Change to “the formation of halo around plaques formed by some bacteriophages”

Done

  • Lines 104-109: These two sentences should be re-written for clarity. A T7 phage was engineered to express DspB in infected cells, which degrades cell-bound exopolysaccharide adhesin. Although the virus targets a single species, this resulted in 100-fold stronger antibiofilm activity in multi-species biofilms relative to non-modified virus.

This fragment modified as follows:
The disruption of the biofilm structure by bacteriophage-encoded enzymes [32] may also contribute to PT efficiency as well as cause POSEs in natural ecosystems. As a proof of this principle  T7 phage was engineered to express DspB in infected cells. Being released during the cell lysis this enzyme degrades the exopolysaccharide (polymeric β-1,6-N-acetyl-d-glucosamine) produced by E. coli TG1 strain. The treatment of the pre-grown E. coli TG1 biofilm with the engineered phage yielded 100 times greater biofilm reduction compared to the non-modified virus [33].

  • Line 152-156. Why do these two sentences constitute a paragraph? It seems like it belongs with the topic in the following paragraph.

Corrected

  • Line 178-180: Re-write this sentence for clarity. I lost the plot starting at “similarly to the own bacterial proteins delivered to the cell surface from the cytoplasm.” Do you mean they perform specific functions that mimic the function of bacterial cell surface proteins?

Exaclty. We had in our mind that own bacterial cell proteins, e.g. adhesins, are made inside the cell and delivered to its surface by secretion systems, while in this case we see phage protein released from different cells due to the lysis that come to the cell surface from without and nevertheless perform functions like the own cell proteins.

 For clarity we modified the sentence as follows:
The bacteriophage proteins released from infected cells may bind to the surface of non-infected cells (or non-induced lysogens) of the same or different species and to functionally mimick the own bacterial surface proteins that are delivered to the cell surface via sercertion pathways.

  • Lines 191-193: run on sentence. Re-write for clarity.

We do not see what’s wrong with this sentence. It points out the question about the possible role of exchange of the surface proteins between bacterial cells via external medium not obligatory due to the phage-mediated effects.

Chapter 3

  • Line 234: Change to “is a 6 a.a. arbitrium peptide…”

corrected

  • Line 235: Change to “When a sufficient concentration of the peptide accumulates…”

corrected

Chapter 4

  • Line 260: Why is this chapter 4 while the above and below sections are labeled as chapter 3?

The revision if performed on the file sent to us by the Editor with her decision letter. In this version we do not see any problems with the chapters numbering.

Chapter 3 (the second chapter 3) – Should this be chapter 5?

  • Renumber all of the sub-sections. The current order is 3.1, 3.3, 3.2, 3.3 (again).

See above (no problem with the numbering present in the file we’re working with)

  • Line 322: change to “with 10 mM ATP…”

corrected

  • Line 329: corrects spelling of “threshold”

Corrected

  • Line 332-350: It is not clear how phage-specific molecules such as base analogs, effect non-infected cells in the population. Make this idea more obvious.

The compounds accumulate in the cell not only in phage DNA but also as free molecules not yet incorporated into the NA. These molecules are released by the lysis and may impact the non-infected cells. The following line was added:

Upon the cell lysis free unusual bases non-incorporated in viral nucleic acids are released to the environment and potentially may influence surrounding cells.

  • Line 376: define OM or spell out “outer membrane.”

Definition is added

  • Lines 368-381: connect the dots more clearly for the reader. Phage-tolerant E. coli cells associated with high Dps à Dps accumulates in outer membrane and there is loss of OmpX à mutants of Dps do not become phage-tolerant à Dps has RNA binding activity à how does this take you to the hypothesis in your last sentence?

The tolerance was most probably due to decrease of OmpX. However, this phenotype did not emerge in dps mutant. In addition to nucleoid binding activity, Dps was shown to bind to RNA. The localization of this protein in the OM linked to the RNA binding allowes O. Ozoline (as referred below) to suggest that Dps in involved in small RNA transfer. Her preliminary results are compatible with this idea, however we are not authorized to present or announce the unpublished research of this group. Therefore, we cite the hypothesis only.

For clarity we added the following lines:

The decrease of OmpX may explain the phage tolerance if this protein or some other protein(s) or structure(s) associated with it, is involved in the reception of the phage. However, the role of the Dps protein is unclear. Interestingly, in a recent study [115], Dps protein was….

  • Line 322: this sentence is awkward due to the phrase “which is especially interesting.” I had to read a few times to unwrap it. Suggestion: In recent years, data has accumulated on the significant role that bacterial non-coding RNAs play in regulating cell metabolism and, interestingly, in the interactions between symbiotic bacteria and their plant and animal macroorganism hosts.

Changed as suggested

  • Line 330: italics for E. coli

Done

  • Line 342: change to “The transfer of small RNAs and other active compunds…”

Done

  • Line 351: typo in the word “also” (alos).

corrected

  • Line 351: Do you mean “latter” here rather than “later?”

Don’t see any “later” here. Could you please cite the context?

  • Line 358-359: incomplete sentence. This process is what or especially what?

Changed for: “The vesicles formatiom  is especially effective in case of phages harboring SAR-type endolysins that degrade peptidoglycan all over the surface of the lysing cell [132].”

  • Line 360-361: Delete phrase “It has to be highlighted” and simply highlight the idea by telling us the idea. Also remove inappropriate preposition “into.” Suggestion to reduce wordiness and odd verb tenses: The EV has many different functions in bacterial physiology including communication within bacterial populations, communication between bacteria and higher organisms, components of biofilm matrix, and nutrient acquisition.

We cannot find any single word “highlighted” over the manuscript.

The phrase changed as follows:
Importantly, the EV have many different functions in bacterial physiology including communication within bacterial populations, communication between bacteria and higher organisms, components of biofilm matrix, and nutrient acquisition. etc. (for a review see [124,133]).

  • Line 375: change preposition to “on which there is fairly extensive literature”

Done

  • Line 382: do you mean “at the moment of selection”? Otherwise it sounds like the cells are resistant to the moment of selection.

Changed for  “….at the moment of stringent selection performed on the plates…”.

  • Line 384-385: what do you mean when you say that hypermutable host variants had a significant advantage over the wild type phage? The ideas in this two-sentence paragraph (which may be too short to communicate a significant idea) are not clearly communicated.  In general, the first sentence should communicate the big idea of the paragraph followed by at least 2-3 sentences that provide evidence supporting that big idea. What is the most important idea about the outcome of low MOI infections that you want to communicate and how can you more specifically illustrate this idea with evidence. In your last sentence of this paragraph you state that “under conditions close to natural conditions, the negative effect of genetic instability seems to outweigh the advantages of better adaptability to the phage….” What are the negative effects of genetic instability (deleterious mutations?) and how do they outweigh advantages? These sentences need more specificity in order to clearly communicate your idea. I don’t really know a lot about this topic but presumably the idea here is that the stress of phage infection results in a hypermutable state of the bacterial host (either due to the phage-encoded recombinases or simply due to stress of infection itself) that lead to mutations provide resistance to phage infection???

The paragraph was re-written to include more detailed explanation of our idea (please refer to the MS)

  • Lines 388-393: Is this the same topic as the previous paragraph (hypermutable host variants) and can this be combined with above paragraph to form a clearer communication of the development of resistance to lytic infection due to hypermutable community members?

Please see above

Conclusions

  • Lines 396-398: awkward and wordy sentence masks the big idea of this sentence. Re-write to clarify. Suggestion: Here we described multiple examples where microbial populations respond to phage infection by modifying the physiology of uninfected bacterial cells. It is not yet clear whether these examples are exotic exceptional cases or are manifestations of more general overlooked principles.

We changed the fragment as it is suggested by the reviewer.

  • Lines 398-399: The development of what kind of screening tools? Fluorescent time lapse microscopy for example?

The idea is discussed below “The reporter systems indicating the activation of the known general stress pathways….”. We do not think that technical discussion of possible detection tools (fluorescet microscopy, flow citometry or other approaches) is necessary for a short conclusion paragraph.